# NOT ALL PARAMETERS ARE EQUAL: A HESSIAN IN­FORMED DIFFERENTIAL LEARNING RATE FOR DEEP LEARNING

## ABSTRACT

Differential learning rate (DLR), a technique that applies different learning rates to different model parameters, has been widely used in deep learning and achieved empirical success via its various forms. For example, parameter-efficient training (PET) applies zero learning rates to most parameters so as to significantly saves the computational cost; adaptive optimizers such as Adam apply the coordinate-wise learning rate to accelerate the convergence.

At the core, DLR leverages the observation that different parameters can have different loss curvature, which is hard to characterize in general. We propose the Hessian-informed differential learning rate (Hi-DLR), an efficient approach that captures the loss curvature of parameters for any model and optimizer adaptively. Given a proper grouping of parameters, we empirically demonstrate that Hi-DLR can improve the convergence by dynamically determining the learning rates during the training. Furthermore, we can quantify the influence of different parameters and freeze the less-contributing parameters, which leads to a new PET that automatically adapts to various tasks and models.

## 1 INTRODUCTION

**DLR by parameter groups** We term the *differential learning rate* (DLR) as a technique that as­signs different learning rates to different parameter groups. Here the parameter groups are partitions of model parameters $\boldsymbol{w} \in \mathbb{R}^D$ such that $\boldsymbol{w} = [\boldsymbol{w}_{(1)}, ..., \boldsymbol{w}_{(K)}]$ and the gradient $\mathbf{g} = [\mathbf{g}_{(1)}, ..., \mathbf{g}_{(K)}]$, and we defer the notations to Section 2.1.

When $K = 1$, this reduces to the uniform learning rate (ULR) and we update with $\eta_t \in \mathbb{R}$ such that

$$\boldsymbol{w}_{t+1} = \boldsymbol{w}_t - \eta_t \mathbf{g}_t.$$

When $K > 1$, we have multiple learning rates $\eta_{(k)}$ for $k \in [K]$ such that

$$\boldsymbol{w}_{t+1} = \boldsymbol{w}_t - [\eta_{(1)}\mathbf{g}_{(1),t}, ..., \eta_{(K)}\mathbf{g}_{(K),t}] := \boldsymbol{w}_t - \boldsymbol{\eta}_{[K],t}\mathbf{g}_{[K],t}$$

**Motivation of DLR.** At high level, DLR can be beneficial because the loss curvature can be very different for different parameters. The loss curvature is captured by the Hessian information matrix and its eigen-spectrum, as demonstrated by Figure 1-5 in Ghorbani (2019), Figure 1 and 3 in Yao et al. (2020), Figure 1 and 6 in Sankar et al. (2021), and Figure 1-2 in Zhang et al. (2024). We visualize in Figure 1 that, by grouping the parameters into biases and weights, the two groups have significantly different curvatures and prefer different learning rates. We give further motivation from an optimization perspective in Section 2.2 and Section 2.3.

**Related works to DLR.** In fact, DLR has been widely used in deep learning, including parameter-efficient training (PET), layer-wise learning rate, and adaptive optimizers.

For example, PET methods include Adapter Houlsby et al., BitFit Zaken et al. (2022), LoRA and its variants Hu et al. (2022); Hayou et al. (2024) and others (see more in Han et al. (2024)) are special cases of two-group DLR, because the majority of parameters is frozen and non-trainable (i.e. using a learning rate of 0) and a small portion of parameters uses a non-zero learning rate. These methods

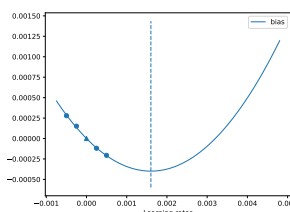 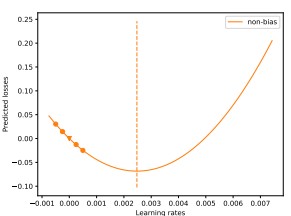 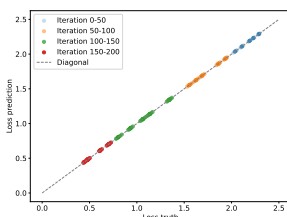

Figure 1: Second-order Taylor approximation in equation 3.1 is sufficiently accurate. We visualize losses with two-group Hi-DLR (bias) under the settings in Section 4.1. Left&Middle: $L(\boldsymbol{w}_{(1)} - \xi_j \mathbf{g}_{(1)})$ and $L(\boldsymbol{w}_{(2)} - \xi_j \mathbf{g}_{(2)})$ in dots at iteration 200. Solid lines are the fitted quadratic functions, with minimizer marked by dashed vertical lines. Right: the loss truth is the left side of equation 3.1 plus $L(\boldsymbol{w})$, and the loss prediction is the right side of equation 3.1 plus $L(\boldsymbol{w})$.

have shown strong performance in fine-tuning large vision and language models, including GPT, ViT, ResNet, etc.

Another example is the layer-wise learning rate. Howard & Ruder (2018) proposed a depth-wise DLR where deeper layers use larger learning rate with $\eta_{(l)} = \eta \cdot 2.6^l$ and $l$ is the layer index. Similar ideas have been proposed not only in fine-tuning but also in pre-training You et al. (2019); Zheng & Kwok (2019); You et al. (2017); Singh et al. (2015); Ginsburg et al. (2019); Zhang et al. (2022); Sun et al. (2019); Ioannou et al. (2023). Some variants have extended the depth-wise DLR in a block-wise manner, by grouping multiple layers into a block for faster computation.

From a different angle, we can view the adaptive optimizers including Adam Kingma & Ba (2014) as SGD with coordinate-wise DLR ($K = D$ where $D$ is the model size), since each parameter uses a different learning rate. For example, we can write SignSGD Bernstein et al. (2018) – a special subcase of Adam as

$$w_{t,i} = w_{t-1,i} - \eta g_{t,i}/|g_{t,i}| = w_{t-1,i} - \eta_i g_{t,i} \tag{1.1}$$

where $\eta_i \equiv \eta/|g_{t,i}|$ is the i-th coordinate's learning rate. Note the general Adam can be written as the SGD with DLR similarly, though the coordinate's learning rate is more complicated due to the exponential moving averages.

Despite the success of DLR in many areas, there are some challenges for its wider application which we introduce in the following.

**Potential challenges in DLR.** Naively implemented, DLR will introduce $K$ hyperparameters instead of 1 hyperparameter by ULR. This leads to the challenge of hyperparameter tuning, which can be prohibitively expensive, especially when model size $D$ or number of learning rates $K$ is large.

One approach that reduces the number of effective hyperparameters in DLR, which is adopted by the aforementioned works, is to incorporate a heuristic structure among $\eta_{(k)}$. As we have discussed, different PET methods freeze different parameters; the depth-wise learning rate uses a fixed ratio 2.6 to scale $\eta_{(l)}$; SignSGD or Adam uses the coordinate-wise gradient norm to scale $\eta_i$, so that $K$ hyperparameters have only 1 degree of freedom in $\eta$.

Nevertheless, such heuristic structure may fail to work in some cases. For example, the fixed ratio in depth-wise learning rate is expected to vary for different models and tasks. We also see in first two rows of Table 2 that LoRA is comparable to full model training (FMT) on SST-2, MNLI and QNLI, but not so on MRPC and ColA. Further experiments in Figure 6 and Figure 7 lend strengths to our observation that no one PET method fits all cases.

We consider an orthogonal approach that preserves the $K$ degrees of freedom and adaptively adjusts $\eta_{(k)}$ with minimal overhead, which can be used in combination with PET and adaptive optimizers.

**Automatic ULR.** In the ULR regime, recent advances have proposed automatic learning rate schedule (or parameter-free, or learning-rate-free methods), including but not limited to D-Adaptation Defazio & Mishchenko (2023), Prodigy Mishchenko & Defazio (2023), DoG Ivgi et al. (2023), and GeN Bu & Xu (2024), among which GeN uniquely leverages the Hessian information.

However, these methods cannot easily optimize the DLR in general. In this work, we extends GeN from ULR to DLR in Algorithm 1. We term our method as *Hessian-informed DLR* (Hi-DLR) and highlight that its subcase Hi-ULR is equivalent to GeN.

We note there are more learning rate techniques that can leverage Hessian information, such as line-search Drori & Taylor (2020); Goujaud et al. (2022); Armijo (1966); Bertsekas (1997), but an extra overhead is incurred for a fine-grained search. We have developed some efficient tricks so that Hi-DLR can be almost as fast as ULR in Section 3.

**Contribution.**

- We introduce Hi-DLR in equation 2.4 to enrich the approximation of Hessian information, in addition to the pre-conditioning of any optimizer, so as to leverage the different loss curvature of different parameters through different learning rates.
- We propose Algorithm 1 to efficiently compute Hi-DLR, with a novel diagonalization trick in equation 3.1, which not only significantly reduces the computation cost, but also separates the contribution of different parameter groups in equation 5.1.
- We demonstrate that Hi-DLR is favorable for various tasks like image/text classification, regression, multi-task learning, as well as parameter-efficient fine-tuning.
- We develop a meta-framework of PET as an application of Hi-DLR, where we use the per-parameter influence to select trainable parameters and thus an adaptive PET method for general models and tasks.

## 2 DIFFERENTIAL LEARNING RATE AND HESSIAN INFORMATION

### 2.1 NOTATIONS

We denote $\boldsymbol{w}$ as the parameters of a model, while $\boldsymbol{w}_t \in \mathbb{R}^D$ represents the iteration $t$ and $\boldsymbol{w}_{(k)}$ represents the $k$-th parameter group. We use $[\boldsymbol{w}_{(1)}, \boldsymbol{w}_{(2)}] \in \mathbb{R}^{m+n}$ to concatenate two parameter groups in $\mathbb{R}^m$ and $\mathbb{R}^n$. The same notation follows for other variables including the mini-batch gradient $\mathbf{g} \in \mathbb{R}^D$, and we denote the learning rates $\boldsymbol{\eta}_{[K]} = [\eta_{(1)}, ..., \eta_{(K)}] \in \mathbb{R}^K$ for $K$ parameter groups. We denote the loss as $L(\boldsymbol{w})$, its first-order derivative as $\mathbf{G}(\boldsymbol{w}) := \frac{\partial L(\boldsymbol{w})}{\partial \boldsymbol{w}}$ and its second-order derivative as $\mathbf{H}(\boldsymbol{w}) := \frac{\partial^2 L(\boldsymbol{w})}{\partial \boldsymbol{w}^2}$. We omit $t$ whenever it is obvious from the context.

### 2.2 A GLOBAL PERSPECTIVE: HYPERPARAMETER OPTIMIZATION PROBLEM OF DLR

To set the stage, we define the hyperparameter optimization problems and give some motivation of using DLR over uniform learning rate (ULR). We aim to minimize the loss after $T$ iterations, given any optimzier and any $K$ groups of the model parameters,

$$\text{ULR problem:} \quad \min_{\eta_{(1)}=...=\eta_{(K)}} L(\boldsymbol{w}_T), \quad \text{DLR problem:} \quad \min_{\eta_{(1)},...,\eta_{(K)}} L(\boldsymbol{w}_T).$$

Here the ULR problem is a univariate optimization, with 1 degree of freedom, which can be solved through grid search, Prodigy, D-adaptation, GeN, etc. In contrast, the DLR problem is a multi-variate optimization, with $K$ degrees of freedom. Therefore, ULR problem is a constrained DLR problem: denoting the optimal learning rate of DLR as $\eta^*_{(1)}, ..., \eta^*_{(K)}$, then the solution of URL problem is sub-optimal unless $\eta^*_{(1)} = ... = \eta^*_{(K)}$.

**Remark 2.1.** Optimizers with coordinate-wise learning rates (e.g. Adam/SignSGD in equation 1.1) are DLR methods, yet the hyperparameter optimization problem is a ULR problem when $K = 1$.

To put this into perspective, we test two functions in Figure 2: (1) the ellipse $L(w_0, w_1) = w_0^2 + 100w_1^2$, which is convex; (2) the sum of Beale and Rosenbrock functions, which is non-convex. We leave more details and explanation in Appendix A.1. We see that our Hi-DLR significantly accelerates the convergence[1] when compared ULR methods.

---

[1]Specifically, for the ellipse function, we note that Hi-DLR reduces to the Newton's method, which is known to find the minimum in one iteration.

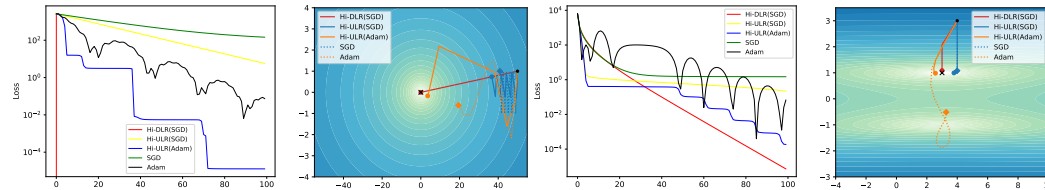

Figure 2: Optimizing over 2D test functions. The left two plots are the results of optimizing an ellipse function; the right two plots show the optimization on a function that is the sum of Beale and Rosenbrock. Hi-DLR is our method; Hi-ULR recovers GeN; the rest uses a manually selected learning rate. See experiment details in Appendix A.1.

## 2.3 A LOCAL PERSPECTIVE: NEXT-LOSS MINIMIZATION

To understand the mechanism of DLR's improvement over ULR, we focus on one iteration and analyze the next-loss minimization along any direction $\boldsymbol{d} \in \mathbb{R}^D$ by $\boldsymbol{w}_{t+1} = \boldsymbol{w}_t - \boldsymbol{d}$. Using the Taylor expansion to capture the loss curvature, we get

$$\min_{\boldsymbol{d}} L(\boldsymbol{w}_{t+1}) = \min_{\boldsymbol{d}} L(\boldsymbol{w}_t - \boldsymbol{d}) \approx \min_{\boldsymbol{d}} L(\boldsymbol{w}_t) - \mathbf{G}_t^\top \boldsymbol{d} + \frac{1}{2} \boldsymbol{d}^\top \mathbf{H}_t \boldsymbol{d} \qquad (2.1)$$

The minimizer $\boldsymbol{d}_t^*$ of equation 2.1 is $\mathbf{H}_t^{-1} \mathbf{G}_t$, which leads to the Newton's method as $\boldsymbol{w}_{t+1} = \boldsymbol{w}_t - \boldsymbol{d}_t^* = \boldsymbol{w}_t - \mathbf{H}_t^{-1} \mathbf{G}_t$.

However, $\mathbf{H}_t \in \mathbb{R}^{D \times D}$ is hard to compute for large-scale optimization, because of the complication in second-order differentiation and the prohibitive memory cost to store $\mathbf{H}_t$. In practice, $\mathbf{H}_t^{-1} \mathbf{G}_t$ is approximated by $\eta_t \mathbf{g}_t^{\text{optim}} \equiv \eta_t \mathbf{P}^{-1} \mathbf{g}_t$, i.e. the pre-conditioned gradient multiplied with a proper learning rate, and thus $\mathbf{H}^{-1} \approx \eta \mathbf{P}^{-1} = (\eta \mathbf{I}) \cdot \mathbf{P}^{-1}$. The majority of existing methods focus on merging the Hessian information into $\mathbf{P}^{-1}$. For example, Adam Kingma & Ba (2014), AdamW Loshchilov & Hutter (2017), AdaGrad Duchi et al. (2011), AdaDelta Zeiler (2012), RMSProp Hinton et al. (2012) use the square root of diagonal Fisher information as $\mathbf{P}^{-1}$; AdaHessian Yao et al. (2021) and Sophia Liu et al. (2023) use the diagonal Hessian information or Gauss-Newton decomposition.

Orthogonal to these works, DLR (with $K$ parameter groups) extends $\eta \mathbf{I}$ to a $K$-dimensional diagonal matrix, up to permutation of elements,

$$\mathbf{H}^{-1} \approx \begin{pmatrix} \eta_{(1)} \mathbf{I} & 0 & ... & 0 \\ 0 & \eta_{(2)} \mathbf{I} & ... & 0 \\ 0 & ... & \ddots & 0 \\ 0 & ... & 0 & \eta_{(K)} \mathbf{I} \end{pmatrix} \mathbf{P}^{-1}$$

As a consequence, DLR enriches the approximation to $\mathbf{H}^{-1}$ with a higher degree of freedom, which allows the learning rates to capture the Hessian information and thus translates to improved convergence when the learning rates $\eta_{(k)}$ are properly set.

## 2.4 OPTIMAL DIFFERENTIAL LEARNING RATES

We now derive the optimal DLR in the sense of equation 2.1 with $\boldsymbol{d} = \boldsymbol{\eta}_{[K]} \mathbf{g}_{[K]}^{\text{optim}}$,

$$L(\boldsymbol{w}_{t+1}) - L(\boldsymbol{w}_t) = -\mathbf{G}^\top (\boldsymbol{\eta}_{[K]} \mathbf{g}_{[K]}) + \frac{1}{2} (\boldsymbol{\eta}_{[K]} \mathbf{g}_{[K]})^\top \mathbf{H} (\boldsymbol{\eta}_{[K]} \mathbf{g}_{[K]}) + o(|\boldsymbol{\eta}_{[K]}|^2) \qquad (2.2)$$

$$\approx -\boldsymbol{\eta}_{[K]}^\top \underbrace{\begin{pmatrix} \mathbf{G}_{(1)}^\top \mathbf{g}_{(1)} \\ ... \\ \mathbf{G}_{(K)}^\top \mathbf{g}_{(K)} \end{pmatrix}}_{\mathbf{b}_*(\mathbf{g}_{[K]}^{\text{optim}}) \in \mathbb{R}^K} + \frac{1}{2} \boldsymbol{\eta}_{[K]}^\top \underbrace{\begin{pmatrix} \mathbf{g}_{(1)}^\top \mathbf{H}_{(11)} \mathbf{g}_{(1)} & \cdots & \mathbf{g}_{(1)}^\top \mathbf{H}_{(1K)} \mathbf{g}_{(K)} \\ ... & ... & ... \\ \mathbf{g}_{(K)}^\top \mathbf{H}_{(K1)} \mathbf{g}_1 & \cdots & \mathbf{g}_{(K)}^\top \mathbf{H}_{(KK)} \mathbf{g}_{(K)} \end{pmatrix}}_{\mathbf{A}_*(\mathbf{g}_{[K]}^{\text{optim}}) \in \mathbb{R}^{K \times K}} \boldsymbol{\eta}_{[K]} \qquad (2.3)$$

This approximation is sufficiently accurate when $\boldsymbol{\eta}_{[K]}$ is small (c.f. Figure 2 in Bu & Xu (2024) when $K = 1$; see also our Figure 1), because the error term $o(\eta^2)$ is very small for the commonly used learning rates.

If $\mathbf{A}_*$ and $\mathbf{b}_*$ are known and if $\mathbf{A}_*$ is positive definite, the quadratic function in equation 2.3 admits a unique minimum at

$$\boldsymbol{\eta}_{\text{Hi-DLR}} = [\eta_{(1)}^*, ..., \eta_{(K)}^*] = \mathbf{A}_*^{-1}\mathbf{b}_* \in \mathbb{R}^K \tag{2.4}$$

which we term as the Hessian-informed DLR (Hi-DLR). Notice that $\mathbf{A}_*$ and $\mathbf{b}_*$ can be defined on any $\mathbf{g}^{\text{optim}}$, hence Hi-DLR applies to any optimizer and the Hessian information is captured by both the pre-conditioning (through $\mathbf{P}^{-1}$ in $\mathbf{g}^{\text{optim}}$) and the learning rate (through $\mathbf{A}_*$ in $\boldsymbol{\eta}_{\text{Hi-DLR}}$). In what follows, we omit the superscript in $\mathbf{g}^{\text{optim}}$ for the simplicity of presentation.

## 3 COMPUTING HI-DLR WITHOUT ADDITIONAL BACK-PROPAGATION

We propose Algorithm 1 to efficiently compute Hi-DLR, which requires the knowledge of $\mathbf{A}_*$ and $\mathbf{b}_*$ in equation 2.3, or equivalently $\mathbf{G}_{(k)}^\top \mathbf{g}_{(k)}$ and $\mathbf{g}_{(k)}^\top \mathbf{H}_{(kk)} \mathbf{g}_{(k)}$. Specifically, we demonstrate what, how, and when to derive these coefficients, thus reducing the computation overhead from $O(D^2)$ to $O(1)$ and allowing Algorithm 1 to be almost as fast as standard optimization. See our detailed complexity analysis in Appendix B.

---

**Algorithm 1** Generalized Newton's optimizers with multiple parameter groups

---

1: **for** $t \in 1, \cdots, T$ **do**
2:      Compute loss $L_0 = L(\boldsymbol{w}_t)$ by the forward pass
3:      Compute gradient $\mathbf{g}(\boldsymbol{w}_t)$ by the back-propagation on $L_0$
4:      Modify gradient as $\mathbf{g} = \mathbf{g}^{\text{optim}}$ by AdamW, momentum SGD, etc.
5:      **if** $t \bmod \Phi == 0$: **then**
6:          **for** $j \in 1, ..., 4K$ **do**:
7:              Randomly select $\hat{\boldsymbol{\eta}} := [\hat{\eta}_{(1)}, ..., \hat{\eta}_{(K)}] \sim N(0, \text{diag}(\boldsymbol{\eta}))$
8:              Compute $L_j = L(\boldsymbol{w}_t - [\hat{\eta}_{(1)}\mathbf{g}_{(1)}, ...])$ by the forward pass
9:          Fit the quadratic function from $\{\hat{\boldsymbol{\eta}}\}_j \to \{L_j - L_0\}$
10:          Derive $\mathbf{G}_{(k)}^\top \mathbf{g}_{(k)}$ and $\mathbf{g}_{(k)}^\top \mathbf{H}_{(kk)} \mathbf{g}_{(k)}$ in equation 3.1
11:          Compute per-parameter influence $\frac{|\mathbf{G}_{(k)}^\top \mathbf{g}_{(k)}|^2}{\mathbf{g}_{(k)}^\top \mathbf{H}_{(kk)} \mathbf{g}_{(k)} \cdot d_k}$ for each group
12:          Derive the optimal learning rate $\boldsymbol{\eta}$ by equation 3.2
13:      Update $\boldsymbol{w}_{t+1} = \boldsymbol{w}_t - [\eta_{(1)}\mathbf{g}_{(1)}, ...]$

---

**What to derive.** $\mathbf{A}_* \in \mathbb{R}^{K \times K}$ contains $O(K^2)$ elements to be derived, which can be costly and hard-to-scale for large $K$ (say $K = 40$ in CelebA), because we will use one forward pass to estimate each element. In practice, we simplify the multivariate quadratic function in equation 2.3 by only deriving the diagonal of $\mathbf{A}_*$,

$$L(\boldsymbol{w}_{t+1}) - L(\boldsymbol{w}_t) \approx -\boldsymbol{\eta}_{[K]}^\top \mathbf{b}_* + \frac{1}{2}\boldsymbol{\eta}_{[K]}^\top \text{diag}(\mathbf{A}_*)\boldsymbol{\eta}_{[K]} = \sum_k (\frac{1}{2}\eta_k^2 \mathbf{g}_{(k)}^\top \mathbf{H}_{(kk)} \mathbf{g}_{(k)} - \eta_k \mathbf{G}_{(k)}^\top \mathbf{g}_{(k)})$$

$$\tag{3.1}$$

which is minimized, if all $\mathbf{g}_{(k)}^\top \mathbf{H}_{(kk)} \mathbf{g}_{(k)}$ are positive, at

$$\eta_k^* = \frac{\mathbf{G}_{(k)}^\top \mathbf{g}_{(k)}}{\mathbf{g}_{(k)}^\top \mathbf{H}_{(kk)} \mathbf{g}_{(k)}} \text{ for } k = 1, ..., K. \tag{3.2}$$

In summary, we derive $\text{diag}(\mathbf{A}_*)$ instead of the full $\mathbf{A}_*$, thus reducing the computation overhead from $O(K^2)$ to $O(1)$ with negligible accuracy degradation empirically.

**How to derive.** We adopt the back-propagation-free approach in Bu & Xu (2024) to fit the quadratic function equation 3.1, without ever instantiating the computationally expensive $\mathbf{G}$ or $\mathbf{H}$. We solve a finite-sum problem:

$$\mathbf{A}_*, \mathbf{b}_* = \arg\min_{A,b} \sum_j |L(\boldsymbol{w}_t - \boldsymbol{\xi}_j \mathbf{g}_{[K]}) - L(\boldsymbol{w}_t) + \boldsymbol{\xi}_j^\top \boldsymbol{b} - \frac{1}{2} \boldsymbol{\xi}_j^\top \boldsymbol{A} \boldsymbol{\xi}_j|^2$$

Note this is a multivariate problem with $2K$ variables and Algorithm 1 uses $4K$ different $\boldsymbol{\xi}_j \in \mathbb{R}^K$.

**When to derive.** We derive $\eta_k^*$ through $\mathbf{A}_*$ and $\mathbf{b}_*$ infrequently, say every $\Phi$ iterations following Bu & Xu (2024). This reduces the overhead from $O(K)$ to $O(1)$ if we set $\Phi = O(K)$. We do not update the learning rate if not all $\mathbf{g}_{(k)}^\top \mathbf{H}_k \mathbf{g}_{(k)}$ are positive, i.e. we use $\boldsymbol{\eta}_{[K]}$ from the previous iteration whenever equation 3.1 is not convex in $\eta_k$.

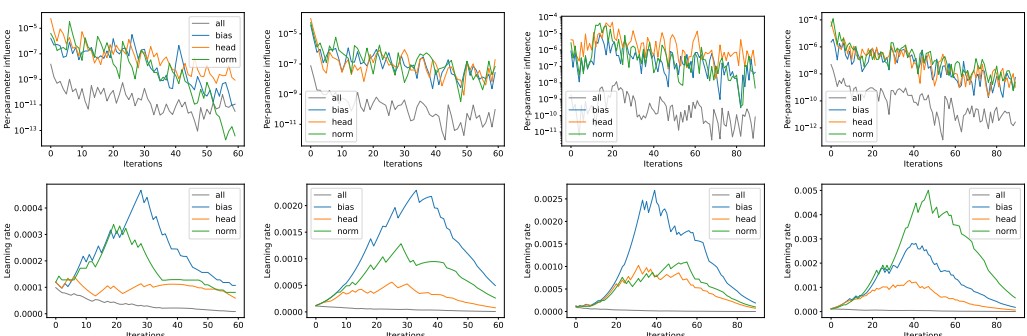

Figure 3: Per-parameter influence in equation 5.1 and learning rates by two-group Hi-DLR under the settings in Section 4.1. Left to right: CIFAR10, CIFAR100, SVHN, and Food101. Note *all* corresponds to Hi-ULR.

## 4 IMPROVING CONVERGENCE WITH HI-DLR

We experiment Hi-DLR with multiple tasks, including image classification, multi-task learning, regression, and language modeling. We leave the experiment details in Appendix A. In what follows, we refer to DLR as the optimization problem with degree of freedom $K > 1$ and ULR otherwise.

### 4.1 IMAGE CLASSIFICATION

We experiment on 5 image datasets for multi-class classification, in which we test 2-group Hi-DLR under full-model fine-tuning. We indicate one parameter group in the parenthesis in Table 1 (e.g. *head*, *bias*, and layer *norm*alization), and treat the remaining parameters as the other group.

Table 1: Test accuracy of ViT (optimized by AdamW) on image classification. We mark the best two results in bold for each dataset.

| Dataset | CIFAR10 | CIFAR100 | Food101 | GTSRB | SVHN |
|---|---|---|---|---|---|
| Reference | Krizhevsky et al. (2009) | | Bossard et al. (2014) | Houben et al. (2013) | Netzer et al. (2011) |
| Hi-DLR (head) | 98.80 | 93.03 | **90.76** | **99.10** | 96.73 |
| Hi-DLR (bias) | **98.95** | **93.40** | **90.68** | **99.07** | 96.80 |
| Hi-DLR (norm) | 98.86 | **93.36** | 90.45 | 99.06 | 96.82 |
| Hi-ULR (GeN) | 98.68 | 92.62 | 90.48 | 99.06 | **97.14** |
| Prodigy | **98.92** | 92.49 | 90.42 | 98.88 | 97.13 |
| D-Adaptation | 97.56 | 88.11 | 89.45 | 99.04 | 96.77 |
| Constant | 97.49 | 89.23 | 88.44 | 98.54 | 96.65 |
| Linear decay | 98.48 | 92.60 | 90.54 | 98.74 | 97.08 |
| Cosine decay | 98.73 | 92.71 | 90.46 | 98.77 | **97.16** |

Widely used ULR methods include heuristic learning rate schedulers (i.e. Constant Raffel et al. (2020), Linear decay Smith (2015) and Cosine decay Loshchilov & Hutter (2016); Radford et al.

(2021)) as well as automatic optimizers like GeN, Prodigy and D-Adaptation. We compare Hi-DLR with these ULR methods and observe that Hi-DLR improves over the best ULR in all datasets except SVHN, since it takes our method some iterations to search the appropriate learning rates.

## 4.2 MULTI-TASK LEARNING

We experiment on CelebA Liu et al. (2015), a large-scale image dataset with 40 labels of face attributes and over 200k samples. This is a multi-label and multi-task problem, each label corresponding to one binary classification task. Hence we have 40 losses in total and will assign 40 learning rates to them. We use a pre-trained ResNet18 He et al. (2016) from Wightman (2019) and only train the last layer, i.e. the classifier head. To be specific, the last layer has a shape $(512, 40)$ and we group the parameters that connect the last hidden layer to each output neuron as one group with shape $(512, 1)$, which corresponds to one task.

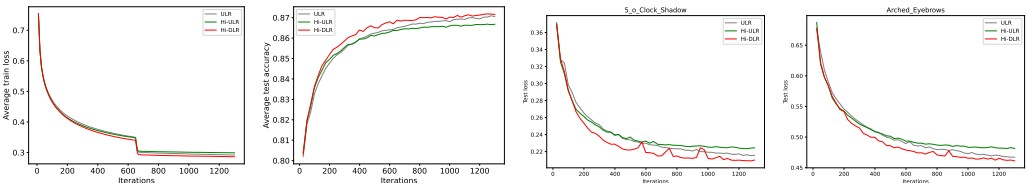

Figure 4: Fine-tuning results on CelebA. From left to right, the first panel shows the average train loss over 40 labels; the second panel shows their average test accuracy; the third and fourth panels are two individual test losses of two labels. See the results of all 40 tasks in Appendix A.2.

In Figure 4 (right two plots), we observe that the difficulty of learning different tasks can vary. Hence assigning different learning rates can improve both overall and individual convergence.

## 4.3 INTERPRETABLE REGRESSION WITH NEURAL ADDITIVE MODEL (NAM)

NAM Agarwal et al. (2021); Xu et al. (2023) is a special neural network architecture, which has multiple sub-networks in parallel such that $g(\boldsymbol{y}) = \beta + \sum_{k=1}^{K} f_j(\mathbf{X_k})$. Here $\boldsymbol{y}$ is the target variable, $g$ is the link function, $\mathbf{X_k}$ is the $k$-th feature of data, $\beta$ is the bias, and $f_k$ is the $k$-th sub-network. Each sub-network attends to a single feature separately so that the effect of each feature is interpretable.

Empirically, different features have various degrees of difficulty in learning, which requires different learning rates during training. We experiment on one synthetic data and the California housing dataset Pace & Barry (1997), as two regression tasks on tabular data. See experiment details in Appendix A.3.

We apply Hi-DLR to $f_k$ as follows: for $K$ sub-networks, we create $K+1$ parameter groups, with one for each $f_k$ and one for the bias $\beta$. The learning rates are shown in the right-most panel of Figure 5. For Hi-DLR, the *lr0* (black lines) is the learning rate for the bias. *lr1, lr2* $\cdots$ is the learning rate selected using Hessian information of parameter group $1, 2, \cdots, K$.

In sum, the experiments in Figure 5 show that NAM with Hi-DLR converges significantly faster than manually selected learning rates or Hi-ULR.

## 4.4 LoRA ON NATURAL LANGUAGE UNDERSTANDING

Low-Rank Adaptation (LoRA, Hu et al. (2022)) is a popular PET method that adds two low-rank matrices to the pretrained weight matrix,

$$\boldsymbol{w} \to \boldsymbol{w} + \boldsymbol{B}\boldsymbol{A}$$

and only trains the parameters in $\boldsymbol{B}$ and $\boldsymbol{A}$. Recent research has shown that freezing $\boldsymbol{A}$ (LoRA-FA, Zhang et al. (2023)) or choosing different learning rates for $\boldsymbol{A}$ and $\boldsymbol{B}$ (Lora+, Hayou et al. (2024)) can boost LoRA's performance. These variants can be deemed as applying DLR to the vanilla LoRA.

We fine-tune RoBERTa-base Liu et al. (2019) model on five GLUE datasets Wang et al. (2018) with LoRA. For Hi-DLR, we split the parameters into three groups: $\boldsymbol{A}$, $\boldsymbol{B}$ and *head*. In Table 2, Hi-DLR

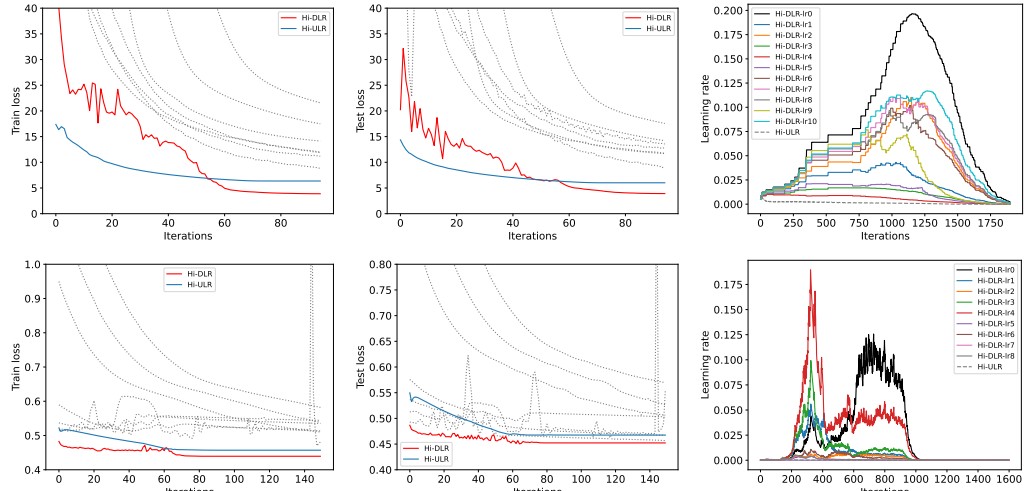

Figure 5: Loss and learning rate of NAM on two regression tasks. The first row is the synthetic dataset. The second row is the California Housing dataset. From left to right, the first two plots show the training losses, and test losses, where the **grey lines** are results trained with a list of manually picked learning rates, the blue curves correspond to Hi-ULR, and the red curves correspond to Hi-DLR; the last plot shows the learning rates for different groups.

Table 2: Performance of RoBERTa-base model with different methods on GLUE datasets. The best performance in PET is marked in bold.

|  | Trainable param | MNLI | SST-2 | MRPC | CoLA | QNLI |
|---|---|---|---|---|---|---|
| ULR (FMT) | 125M | 87.45 | 94.38 | 88.97 | 80.82 | 92.46 |
| ULR (PET) | 0.3M | 85.01 | 93.81 | 75.49 | 69.13 | **91.05** |
| Hi-ULR (PET) | 0.3M | 82.49 | 93.35 | 83.58 | 79.58 | 90.43 |
| Hi-DLR (PET) | 0.3M | **85.21** | **94.15** | **85.78** | **81.59** | 90.48 |

outperforms Hi-ULR and ULR in PET on 4 out of 5 datasets. Experiment details can be found in Appendix A.4.

We notice that LoRA can underperform FMT significantly on some datasets such as CoLA and MRPC. This phenomenon has also been witnessed in other models (see Table 1 of Wang et al. (2024); Wang & Liang (2024)). Additionally, Table 4 of Bu & Xu (2024) shows that BitFit Zaken et al. (2022), another PET method can outperform LoRA one some GLUE datasets but not on others.

These evidences indicate that there is no one PET that can fit all tasks, which is further confirmed in the next section and motivates our new PET method.

## 5 HESSIAN-INFORMED INFLUENCE OF PARAMETERS

In this section, we leverage DLR to quantify the influence of parameters, which identifies the important parameters that could lead to new PET.

### 5.1 PER-PARAMETER INFLUENCE

From equation 3.1 and under the Hi-DLR $\eta_k = \eta_k^*$ in equation 3.2, we can attribute the loss improvement to each parameter group: we can separate the total improvement $\sum_k \frac{|\mathbf{G}_{(k)}^\top \mathbf{g}_{(k)}|^2}{\mathbf{g}_{(k)}^\top \mathbf{H}_{(kk)} \mathbf{g}_{(k)}}$ so that each summand is the group's contribution, and we define

$$\text{Per-Parameter Influence (PPI)} = \frac{|\mathbf{G}_{(k)}^\top \mathbf{g}_{(k)}|^2}{\mathbf{g}_{(k)}^\top \mathbf{H}_{(kk)} \mathbf{g}_{(k)} \cdot d_k} \quad (5.1)$$

where $d_k$ is the number of parameters in group $k$ that sums to $\sum_k d_k = D$. Note the PPI is computed during training by Algorithm 1.

We visualize the PPI in Figure 3 for $K = 2$ and image classification. We further visualize in Figure 6 and Figure 7 for $K \geq 5$ across CV, NLU, NLG tasks. Here we have equipped a model with parameter groups in LoRA (Hu et al. (2022); with module names *lora_A* and *lora_B*), BitFiT (Zaken et al. (2022); *bias*), linear probing (*head*), LayerNorm tuning (Zhao et al.; *norm*), and embedding layer tuning (*embed*).

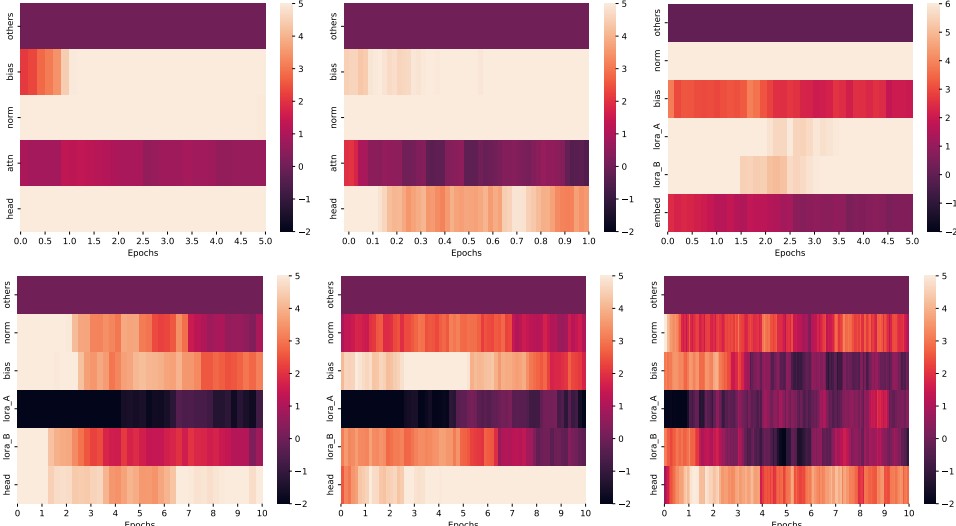

Figure 6: Heatmap of PPI for multiple parameter groups in log-scale. Upper row, left to right: (CIFAR100,ViT-base), (ImageNet, ViT-base), (E2E, GPT2). Lower row, left to right: (MRPC,RoBERTa-base), (CoLA,RoBERTa-base), (SST-2, RoBERTa-base).

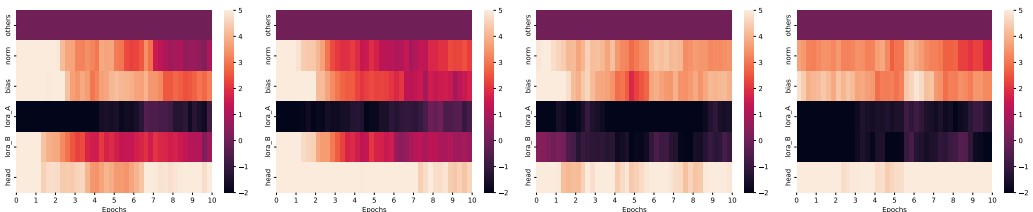

Figure 7: Heatmap of PPI on CoLA dataset in log-scale. Left to right: RoBERTa-base, RoBERTa-large, T5-small, and T5-base.

We consistently observe that existing PET methods indeed have selected the highly influencial parameters, which have about $10^4\times$ higher PPI than the majority of model parameters in Figure 3. This supports the effectiveness of PET, in the sense that it may suffice to train with a small portion of selected parameters and freeze most parameters, with little degradation in performance.

The light-colored regions in Figure 6 have revealed some PET methods, which may be new. On CIFAR100 and ImageNet, ViT model can be effectively trained with BitFit together with LayerNorm tuning; on E2E, GPT2 model can leverage LoRA together with LayerNorm tuning; on RoBERTa models, LoRA with frozen A (LoRA-FA Zhang et al. (2023)) and BitFit seem to work well. It is clear that different datasets can lead to different PPI even on the same model, e.g. the last row in Figure 6, and so can different model architectures, e.g. T5 v.s. RoBERTa in Figure 7.

In summary, we have obtained that there is no one PET method that fits all cases, and the PPI is highly dependent on the tasks (i.e. model architectures, datasets and parameter groups). Specifically, a combination of multiple PET methods usually gives the optimal performance[2]. In what follows, we propose a meta-framework that adaptively identifies strong PET methods given any task.

---

[2]For example, the LoRA library Hu et al. (2022) states that *training bias vectors in tandem with LoRA might be a cost-efficient way to squeeze out extra task performance*.

## 5.2 A META-FRAMEWORK OF ADAPTIVE PET

Our meta-framework is flexible and model-agnostic: given a number of PET methods and the corresponding parameter groups, we leverage Algorithm 1 to select the parameter groups with high PPI and freeze the others if $\text{PPI}_k < \psi \cdot \min_k \text{PPI}_k$. Here $\psi$ is an adjustable hyperparameter, with $\psi = 1$ meaning full model training (FMT) and $\psi > 1$ meaning PET. We note that higher $\psi$ leads to fewer trainable parameters and likely worse performance but better computation efficiency).

In particular, we can determine $\psi$ and thus the PET method by experimenting on a small model, and then transfer to larger models, since we empirically observe that different model sizes (within the same architecture) have similar PPI by parameter groups in Figure 7.

Table 3: Performance of RoBERTa models on CoLA. (Y)es indicates a parameter group is trainable. (N)o indicates a group is frozen. We transfer the PET identified at $\psi = 10$ to larger model.

| model | RoBERTa-base | | | | | | RoBERTa-large | |
|---|---|---|---|---|---|---|---|---|
| $\psi$ | 1 | 1.1 | 10 | 500 | 1000 | 2000 | FMT | PET |
| others | Y | N | N | N | N | N | Y | N |
| norm | Y | Y | Y | N | N | N | Y | Y |
| bias | Y | Y | Y | Y | Y | N | Y | Y |
| head | Y | Y | Y | Y | Y | Y | Y | Y |
| loraA | Y | Y | N | N | N | N | Y | N |
| loraB | Y | Y | Y | Y | N | N | Y | Y |
| accuracy | 84.37 | 81.97 | 82.16 | 81.88 | 81.88 | 80.82 | 85.71 | 84.66 |
| num param | 124.94 | 1.00 | 0.86 | 0.84 | 0.69 | 0.59 | 356.14 | 1.76 |
| % param | 100 | 0.80 | 0.69 | 0.67 | 0.55 | 0.47 | 100 | 0.49 |

Table 4: Performance of GPT models on E2E. (Y)es indicates a parameter group is trainable. (N)o indicates a group is frozen. We transfer the PET identified at $\psi = 10$ to larger models.

| model | GPT2-small | | | | | GPT2-medium | | GPT2-large | |
|---|---|---|---|---|---|---|---|---|---|
| $\psi$ | 1 | 1.1 | 10 | 500 | 1000 | FMT | PET | FMT | PET |
| Others | Y | N | N | N | N | Y | N | Y | N |
| norm | Y | Y | Y | Y | Y | Y | Y | Y | Y |
| bias | Y | Y | N | N | N | Y | N | Y | N |
| loraA | Y | Y | Y | N | N | Y | Y | Y | Y |
| loraB | Y | Y | Y | Y | N | Y | Y | Y | Y |
| embed | Y | Y | N | N | N | Y | N | Y | N |
| perplexity | 3.09 | 3.15 | 3.43 | 3.61 | 3.79 | 3.02 | 3.26 | 2.96 | 3.12 |
| num param | 124.58 | 39.65 | 0.18 | 0.11 | 0.04 | 355.21 | 0.49 | 774.76 | 0.92 |
| % param | 100 | 31.82 | 0.15 | 0.09 | 0.03 | 100 | 0.14 | 100 | 0.12 |

In Table 3 and Table 4, we first experiment on the smaller models, RoBERTa-base and GPT2-small. We allocate 10% of training iterations to determine the PET method at each indicator ranging from 1.1 (training any PET parameters that are more influential than the majority) to 1000 (beyond which all parameters are frozen). We observe that the model performance tend to worsen as $\psi$ increases and the percentage of trainable parameters quickly drops below 1%. We then transfer the PET method at $\psi = 10$ to larger models, which enjoy $\approx 150\%$ training speed and similar performance compared to FMT even though the trainable parameters is $< 0.5\%$ of the full model.

## 6 DISCUSSION

In this work, we have demonstrated that different parameters have different loss curvatures and influences on the convergence, through the lens of Hi-DLR. We propose an efficient algorithm to compute Hi-DLR adaptively, so as to leverage it for faster convergence or PET strategies. We believe there are more and new ways to leverage Hessian information from DLR, that could be future directions. We also leave a discussion of Hi-DLR's limitations in Appendix C.

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

## A EXPERIMENT DETAILS

### A.1 TOY DATA FOR OPTIMIZATION

To manually select the best learning rate, we grid search from $\{1e - 5 * 10^{k/2}\}$ for $k = 0, ..., 11$. The learning rate that gives the smallest loss after 100 iterations will be chosen.

**Ellipse function**

$$\text{Ellipse}(w_0, w_1) = x^2 + 100y^2$$

. We optimize from the initialization at $(w_0, w_1) = (50, 1)$. The minimizer of the ellipse function is $(w_0, w_1) = (0, 0)$.

**Sum of Beale and Rosenbrock** Beale is a convex function and Rosenbrock is a non-convex function.

$$\text{Beale}(x, y) = (1.5 - x + xy)^2 + (2.25 - x + xy^2)^2 + (2.625 - x + xy^3)^2$$
$$\text{Rosenbrock}(x, y) = 100(y - x^2)^2 + (1 - x)^2$$

The unique minimizer for Beale is $(3, 0.5)$, for Rosenbrock is $(1, 1)$. The optimization problem is a sum of Beale and Rosenbrock:

$$L(w_0, w_1) = \text{Beale}(w_0, 0.5) + \text{Rosenbrock}(w_1, 1).$$

So the minimizer of this new $L$ is $(3, 1)$. We optimize from the initialization at $(4, 3)$.

### A.2 MULTI-TASK LEARNING ON CELEBA

Each result is trained on 2 epochs with a training batch size of 500, optimized by a standard AdamW optimizer. No data augmentation is used. For ULR, we use a fixed learning rate of 1e-3. For Hi-ULR and Hi-DLR, we use an initial learning rate 1e-3 and $\Phi = 10$.

### A.3 INTERPRETABLE REGRESSION WITH NAM

**Synthetic data** The data $\boldsymbol{X} \in \mathbb{R}^{3000 \times 10}$. Let's denote the $j$-th column of $\boldsymbol{X}$ as $\boldsymbol{X_j}$. $\boldsymbol{y}$ is generated by an additive model:

$$\boldsymbol{y} = \sum_{i=1}^{10} f_i(\boldsymbol{X_i}) + \mathcal{N}(0, 1)$$

where $f_j$ are zero functions for $j = 7, 8, 9, 10$. The rest features are generated in the following way:

$$f_1(x) = 2x^2 \tanh x, \quad f_2(x) = \sin x \cos x + x^2, \quad f_3(x) = 20/(1 + e^{-5 \sin x})$$
$$f_4(x) = 20 \sin^3 2x - 6 \cos x + x^2, \quad f_5(x) = x^3, \quad f_6(x) = x$$

For synthetic regression data, learning rates for ULR are selected from the list [5e-4, 7e-4, 1e-3, 3e-3, 5e-3, 7e-3, 1e-2]. All the models are trained with SGD. The total number of epochs is 100 and batch size is 256. $\Phi = 2$ for Hi-ULR and Hi-DLR. Plots start from the 5th epoch.

**California housing** This dataset collects the house values of various California districts in 1990. The regression task is to predict house prices with 20,640 examples and 8 housing features including location, layout, etc.

For California housing, learning rates for ULR are selected from a list [5e-6, 7e-6, 1e-5, 3e-5, 5e-5, 7e-5, 1e-4]. We use the Adam optimizer. The total number of epochs 200 is and batch size is 256. $\Phi = 8$ for Hi-ULR and Hi-DLR. Plots start from the 50th epoch.

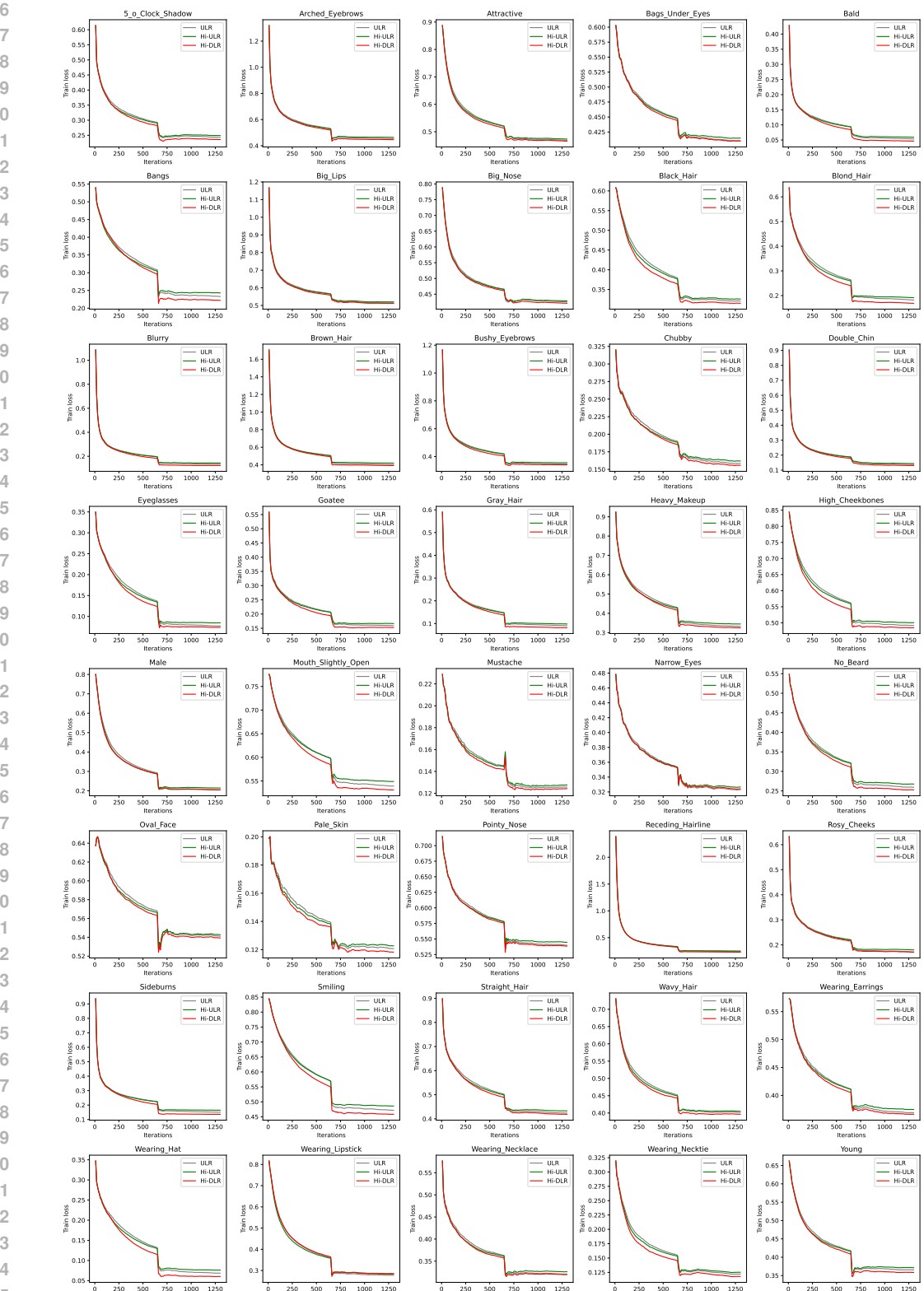

Figure 8: Individual train loss for 40 different labels of fine-tuning CelebA.

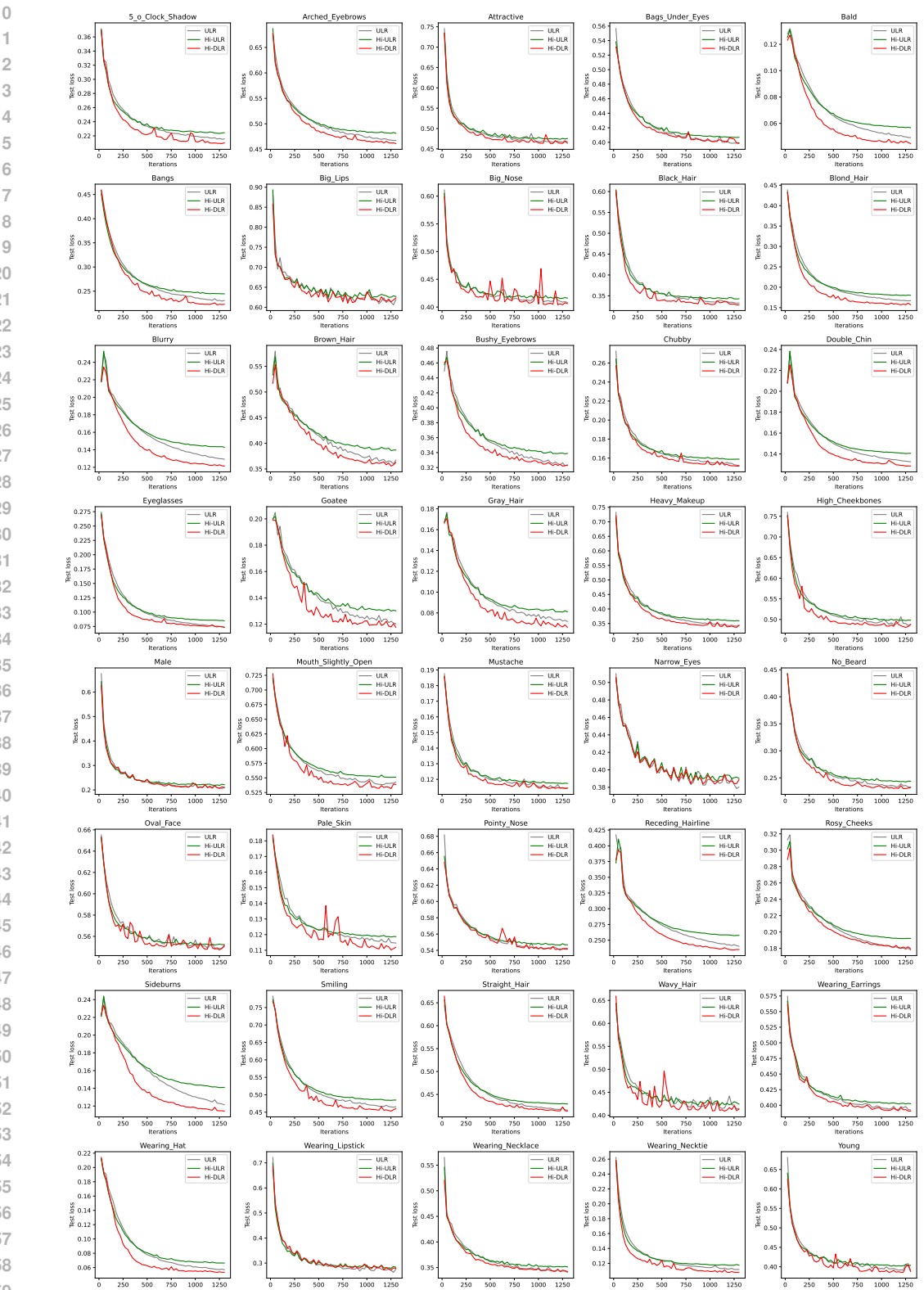

Figure 9: Individual test loss for 40 different labels of fine-tuning CelebA.

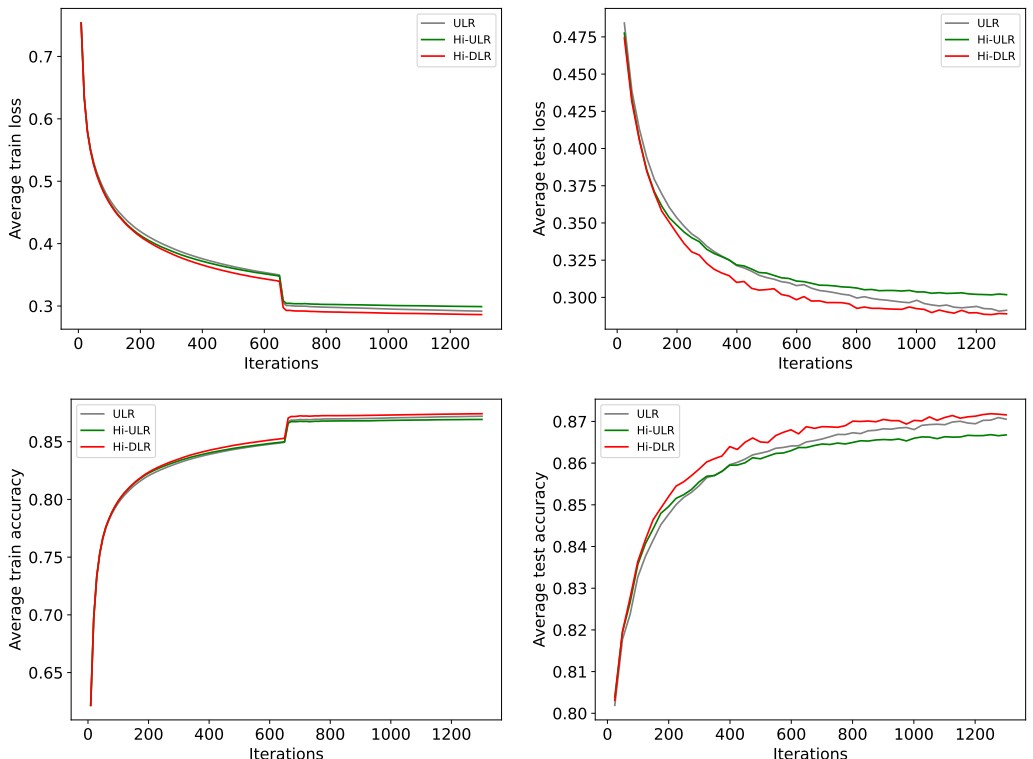

Figure 10: Average performance of fine-tuning CelebA over 40 labels.

## A.4 LoRA on natural language understanding

**Synthetic data**   Except for the GLUE benchmarks, we also experimented with a toy example in LoRA+ to better demonstrate DLR's power. The settings are the same as it is in Appendix C.1.1. of Hayou et al. (2024) except for $n$. We use $n = 1000$ instead of $n = 100$.

We train on 1000 iterations for each method and the plots start from the 50th epoch. For ULR, we grid search for the best learning rate based on the last test loss after 500 iterations. Assume $\eta_A$ and $\eta_B$ is the learning rate for $A$ and $B$ respectively. The search range for $\eta_A$ is $10^k$ for $k$ evenly searched from -4 to -3 for 20 points. The $\eta_B$'s search range starts from $k = -4$ to $k = -1$ for 20 points.

Finally, the selected ULR learning rates are $(\eta_A, \eta_B) = (1e - 4, 1e - 4)$. The best DLR learning rate are $(\eta_A, \eta_B) = (1e - 4, 1e - 1)$.

**NLU tasks**   For NLU tasks, we use batch size 128 for all datasets. The evaluation metric is test accuracy. We use AdamW with a Cosine scheduler and warm-up ratio of 0.03. For every dataset, the full fine-tuning learning rates are 10 times smaller than their corresponding LoRA learning rate. The lazy frequency is selected based on batch size and data size.

| | Data size | Initial learning rate for FT | # of epochs | $\Phi$ |
|---|---|---|---|---|
| MRPC | 3668 | 4e-5 | 3 | 4 |
| SST2 | 67349 | 5e-5 | 3 | 10 |
| MNLI | 392702 | 5e-5 | 1 | 10 |
| CoLA | 8551 | 4e-5 | 1 | 1 |
| QNLI | 104743 | 4e-5 | 3 | 10 |

Table 5: Hyper-parameters for GLUE training.

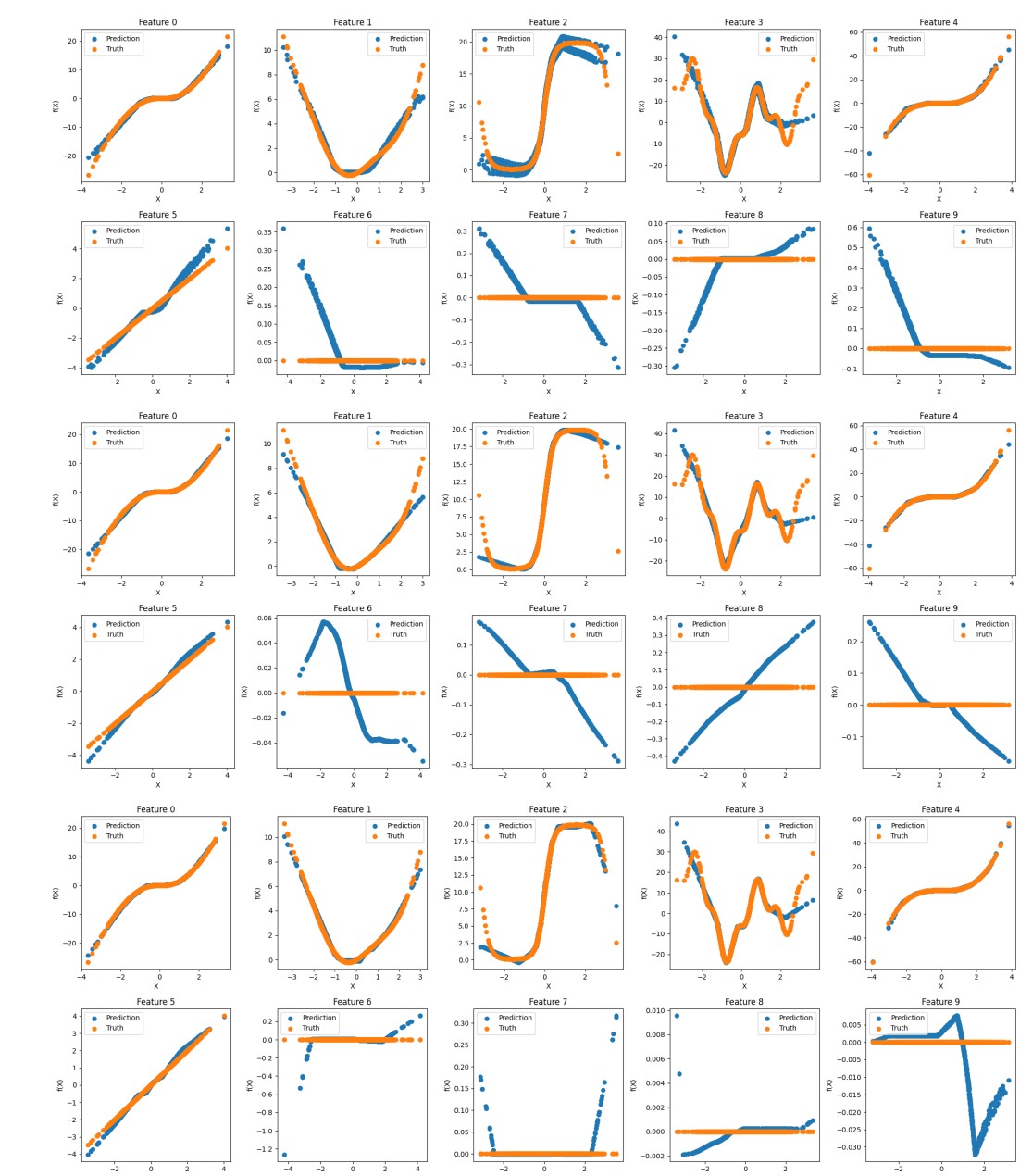

Figure 11: Individual effect learned by NAM on synthetic regression. Blue dots are predictions and orange dots are the truth. The first two rows are results optimized by ULR. The next two rows are features learned by Hi-ULR. The last two rows are the results of Hi-DLR.

For hyper-parameters not mentioned here, we follow Table 9 of Hu et al. (2022).

### A.5   GPT2

For GPT2, we experimented on the E2E dataset. The initial learning rate for full fine-tuning is 1e-4 while it is 1e-3 for PET. The sequence length is 128, the total batch size is 256 and the total validation batch size is 64. The total number of epochs for GPT2-small is 5, and for GPT2-medium and large is 3. The rest hyper-parameters are the same as in Hu et al. (2022).

## A.6 ViT CLASSIFICATION

We use the pre-trained `ViT-base-patch16-224` which can be can be loaded from `timm` library. This model has been trained on ImageNet following Dosovitskiy et al. (2020). We resize all images to 224x224 and normalize the pixel values to [-1,1]. We use AdamW optimizer with the default hyperparameters in Pytorch, except the learning rates. For methods that are not ours, we follow the learning rate settings in Bu & Xu (2024). For Hi-DLR, we use initial learning rate 1e-4, which is the same as Hi-ULR (GeN). We use batch size 500 across datasets with $\Phi = 4$.

## B COMPLEXITY ANALYSIS

We follow the same analysis as in Bu & Xu (2024) and it follows that Hi-DLR has the same peak memory cost as a base optimizer. For time complexity, we consider three operations: the forward pass $F$, the back-propagation $B$ and other costs $C$. Therefore, the base optimizer takes $F + B + C$ whereas Hi-DLR takes $(1 + \frac{4K}{\Phi})F + B + C$. Here the additional computation is from extra forward passes. In a full-parameter training on a single GPU, $C$ is negligible and $B \approx 2F$, the relative training speed of Hi-DLR is $\frac{1}{1+\frac{4K}{3\Phi}}$. For instance, when $K = 3, \Phi = 10$, Hi-DLR is roughly 70% as fast as a base optimizer. While training with PET methods, the $B \approx F$, the relative speed becomes $\frac{1}{1+\frac{4K}{2\Phi}}$. When $K = 3, \Phi = 10$, Hi-DLR is roughly 62.5% as fast as a base optimizer.

## C LIMITATIONS

The success of DLR depends on the grouping of parameters: a sub-optimal grouping strategy might lead to a less effective learning rate adaptation. It remains an interesting future direction on how to leverage human's prior knowledge to efficiently find a good grouping strategy. Computation-wise, the training time of Hi-DLR increases linearly with the number of groups $K$ unless $\Phi$ also increases linearly, limiting its application to very large $K$ if the total number of iterations is small.

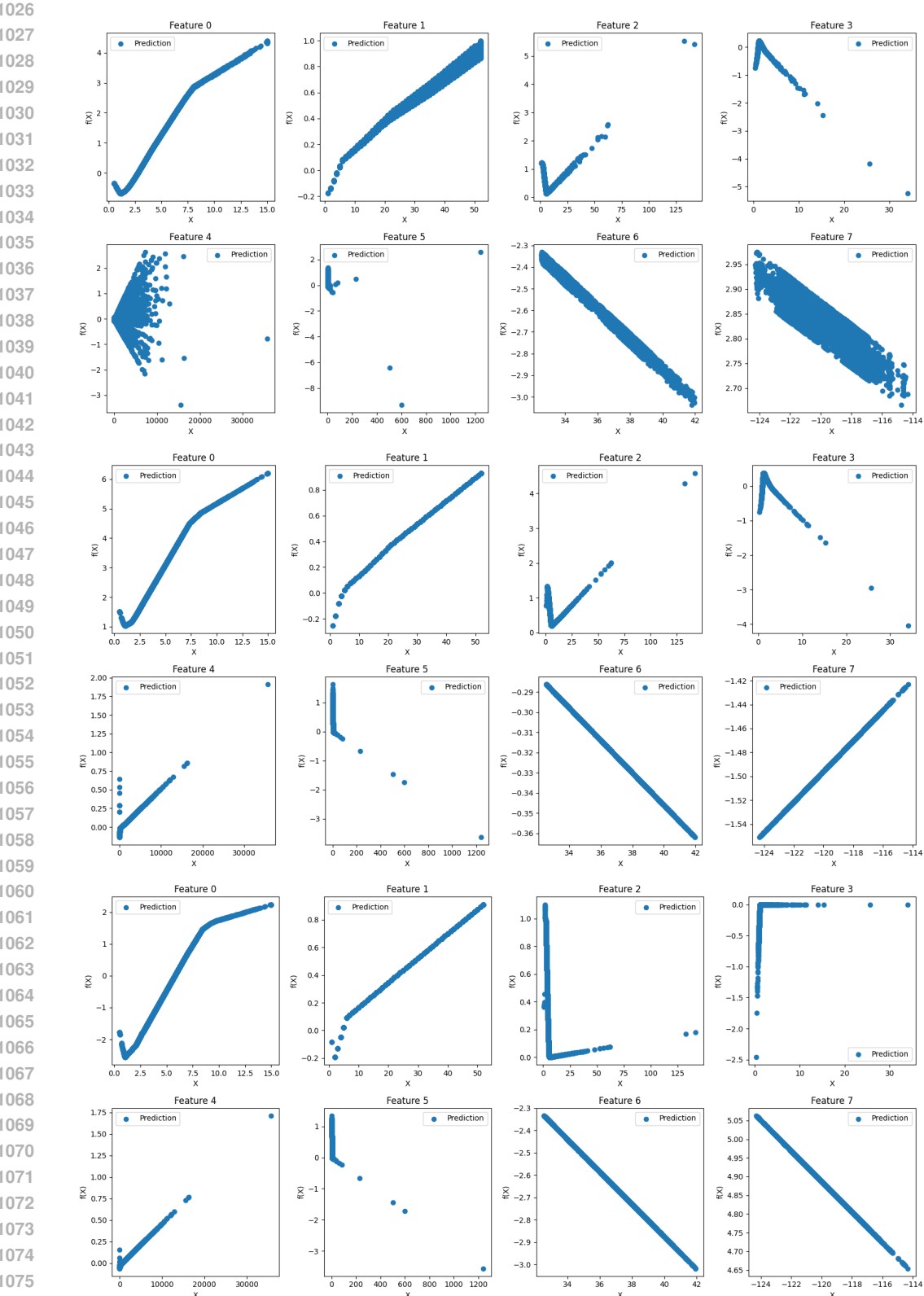

Figure 12: Individual effect learned by NAM on California housing data. Blue dots are predictions. The first two rows are predictions of NAM optimized by ULR. The next two rows are features learned by Hi-ULR. The last two rows are the results of Hi-DLR.

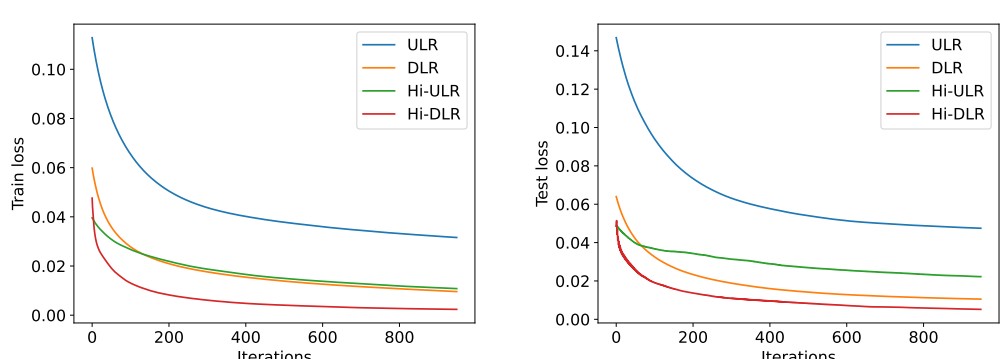

Figure 13: LoraPlus Synthetic data

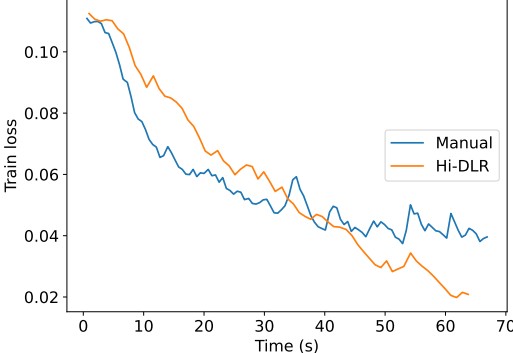

Figure 14: The loss of Hi-DLR ($K = 3, \Phi = 10$) v.s. Cosine decay learning rate on RoBERTa-base on CoLA. The x-axis is the wall-clock training time on a single L4 GPU. The experiment details are the same as in Appendix A.4.

