# OpenReview forum: "Not all parameters are equal: a Hessian informed differential learning rate for deep learning"
_ICLR.cc/2025/Conference — ICLR 2025 Conference Withdrawn Submission_

### Official Review · Reviewer_gPHg · 2024-11-02

**Soundness:** 2
**Presentation:** 3
**Contribution:** 2
**Rating:** 5
**Confidence:** 3

**Summary:**

This paper proposes a differential learning rate strategy called Hi-DLR. The algorithmic novelty of Hi-DLR lies in diagonalizing the Hessian matrix $H$ and extending the first-order approximation of $\mathbf{G}_t^{\top} \mathbf{g}_t^{\text{optim}} / \left(\mathbf{g}_t^{\text{optim}}\right)^{\top} \mathbf{H}_t \mathbf{g}_t^{\text{optim}}$ from [1] from ULR to Differential Learning Rate (DLR). The authors also adopt a per-parameter influence derived from Hi-ULR to select influential parameters for parameter-efficient training. From the reviewer's perspective, this definition of parameter influence is novel, although it is the optimal solution of equation 3.1 with a diagonalized version of the Hessian matrix. The reviewers welcome discussions with other reviewers and the Area Chair to determine if this definition of influence demonstrates conceptual novelty.

[1] Automatic gradient descent with generalized Newton’s method. arXiv, 2024.

**Strengths:**

- The paper is well-motivated and designed to create suitable learning rates for different parameters.
- The authors also provide applications of the proposed Hi-DLR to NAM and LoRA.
- The per-parameter influence metric is interesting.

**Weaknesses:**

**Evaluation of Hi-DLR**: All training loss figures in this paper are plotted with respect to iterations (e.g., Figures 4 and 5). Could the authors provide wall-clock time comparisons between Hi-DLR and baseline methods, in addition to the iteration-based plots. This would help demonstrate whether Hi-DLR provides real-world speedups.

**Questions:**

The authors state that they consistently observe existing PET methods selecting highly influential parameters, which have approximately ($10^4 \times$ ) higher PPI than the majority of model parameters. Why is this the case? The PPI metric is an estimation of parameter influence, so what is the corresponding ground truth for parameter influence? Without ground truth, how can we be certain that PET methods have indeed selected the highly influential parameters?

---

> ### Author Response · Authors · 2024-11-22
>
> We thank the reviewer for the comments! We address them below and welcome more feedbacks. We would appreciate it if the reviewer could raise the score if satisfied.
>
> *Evaluation of Hi-DLR*
>
> Thank you for mentioning this. We have added a detailed complexity analysis in Appendix B in our revised version. In a full-parameter training on a single GPU, the relative training time of Hi-DLR is $\frac{1}{1+\frac{4K}{3\Phi}}$. For instance when $K=3,\Phi=10$, Hi-DLR is 70\% as fast as a base optimizer.
>
> *The authors state that they consistently observe existing PET methods selecting highly influential parameters, ... Why is this the case?
> The PPI metric is an estimation of parameter influence, so what is the corresponding ground truth for parameter influence? Without ground truth, how can we be certain that PET methods have indeed selected the highly influential parameters?*
>
> It remains an open question (that may be out of the scope of this paper) why certain PET methods are effective and select highly influential parameters. Some theories have reasoned that the fine-tuning information is low-rank so a small portion of trainable model parameters can capture it. We are happy to include some references if the reviewer is interested. In deep learning, unfortunately we don't have the ground truth of the true parameter influence. Our approach is to first identify the highly influential parameters through our PPI metric, then determine which PET methods include these parameters. E.g. in Figure 7 third sub-plot, LoRA includes such parameters but BitFit does not. We then validate our determination on the small models by directly trying it on larger models (see Table 3,4), and if the new PET is effective (i.e. comparable to the full fine-tuning), then we are more confident about the selection.

---

> > ### Comment · Reviewer_gPHg · 2024-11-23
> >
> > Thank you for your response.
> >
> > Regarding the evaluation of Hi-DLR, the results in Table 1 suggest that the performance gain brought by Hi-DLR is relatively limited. This raises the question of whether the 40% (i.e., 1/(1-0.3)-1) increase in training time is justified. To better understand and evaluate the practical benefits of Hi-DLR, it would be very helpful to see how the accuracy progresses with respect to wall-clock time, rather than just iterations. This comparison would provide a clearer picture of the trade-offs in real-world scenarios.
> >
> > The reviewer plans to maintain the score for now and will discuss further with the other reviewers during the discussion phase before finalizing the evaluation.

---

> > > ### Author Response · Authors · 2024-11-24
> > >
> > > Thanks for your quick response. We added Figure 14 that compared Hi-DLR's performance with a cosine decay ULR (which cannot be easily tuned as it requires grid search for the peak lr) in wall-clock training time. We observe that the training time overhead is 15\% (would vary in different models/datasets), which is well-characterized by our new complexity analysis in Appendix B (which states the upper bound of overhead to be 16%). Additionally, we kindly remind that even if Hi-DLR has a similar performance than the optimal ULR, it allows an additional advantage of parameter importance that has been highlighted in Sec 5, without extra overhead.
> > >
> > > We hope the reviewer can consider raising the score or let us know how we can further improve!

---

> > > > ### Comment · Reviewer_gPHg · 2024-11-25
> > > >
> > > > Thanks. The reviewer will take this into final consideration.

---

### Official Review · Reviewer_dT3U · 2024-11-02

**Soundness:** 2
**Presentation:** 2
**Contribution:** 2
**Rating:** 3
**Confidence:** 4

**Summary:**

This paper proposes Hessian informed differential learning rates for deep learning, which is derived based on the second-order Taylor approximation of the loss function. Experiments on synthetic data and real data are conducted to demonstrate the effectiveness of the proposed learning rates.

**Strengths:**

1. The authors conducted extensive experiments covering multiple different settings, from synthetic data and toy objective functions to various advanced practical learning tasks.

**Weaknesses:**

1. The novelty of the proposed method is questionable. The idea is very similar to the derivation of Newton’s method, and the approximation of the Hessian with the diagonal matrix is not novel either, for example

Andrei, N. A diagonal quasi-Newton updating method for unconstrained optimization. Numer Algor 81, 575–590 (2019). https://doi.org/10.1007/s11075-018-0562-7

2. The logic of the proposed method is not convincing. The authors mentioned in the caption of Table 1 that they apply the proposed learning rates to AdamW. However, the proposed learning rates are derived based on equations (2.1) and (2.3), where the vector $\mathbf{d}$ in (2.1) is replaced by  $ \boldsymbol{\eta}\_{[K]} \mathbf{g}\_{[K]}^{\mathrm{optim}} $. If the learning rates are applied to AdamW, why shouldn’t we set the vector $\mathbf{d}$ in (2.1) to be the actuarial difference between iterates of AdamW, taking the entry-wise adaptive learning rates of AdamW into consideration? I think the notation $ \mathbf{g}\_{[K]}^{\mathrm{optim}} $ is not explained well either. I suggest that the authors should clarify how exactly their method integrates with AdamW.

3. The paper does not provide sufficient experimental details. Although some are provided in the appendix, the information provided are insufficient to reproduce the results. For example, I do not find any experimental details about the training of the ViT. For example, did the authors consider data augmentation? What is the batch size for training ViT, and what exact version of ViT is used on the various data sets? (e.g., what is the classifier, what are the dimensions of heads, is dropout used, how many patches are each image split into, etc). The authors provide no codes either.

4. For relatively large models, as far as I can tell, the paper only considers applying the proposed learning rates for fine-tuning. It is important to also demonstrate the performance of the proposed method in pre-training. Therefore, I suggest that the authors should include experiments on pre-training certain models such as ViTs.

5. The presentation of the paper is not clear enough, and the paper needs some significant revision. As mentioned in the points above, there are notations and experimental setups that are not explained clearly. Moreover, I feel that this paper lacks a proper introduction section.

**Questions:**

I suggest that the authors should respond to the weaknesses pointed out above.

---

> ### Author Response · Authors · 2024-11-21
>
> We thank the reviewer for all the constructive feedback and the well-grounded questions. We will address your comments one by one. We would sincerely appreciate it if the reviewer could provide more feedback or questions!
>
> 1.*"The idea is very similar to the derivation of Newton’s method"*
>
> we have built the connection between our method and Newton in line 179-181 but also stated the differences in line 182-190. The Newton’s method requires the inverse of Hessian, which is hard to compute in large-scale model trainings.
>
> *"the approximation of the Hessian with the diagonal matrix is not novel either"*
>
> We would like to stress that we’re not diagonalizing Hessian/Fisher. We have stated many classic methods like Adam that diagonalize Hessian/Fisher by introducing preconditionings in line 182-190. As a comparison, our diagonalization trick is applied to gHg, which is an important componant of our Hessian-informed adaptive learning rate. Hence it can be combined with any general optimizer that may or may not use the diagonalization of Hessian/Fisher. We provide the following simplified formula to highlight the difference between our framework and others. (Equation 2.2, 2.3 give a more sophisticated expression of our diagonalization trick.)
>
> SGD: $\eta * I$
>
> Newton: $1*H^{-1}$
>
> AdaHessian: $\eta*diag(H)^{-1}$
>
> Adagrad/Adam: $\eta*diag(P)^{-1}$
>
> ———————————————————————————
>
> Hi-DLR SGD: $\frac{Gg}{ diag(gHg)}*I$
>
> Hi-DLR Adam: $\frac{Gg}{diag(gHg)}*diag(P)^{-1}$
>
>
> 2. From the mathematic viewpoint, our implementation on AdamW is $Gg/diag(gHg)*diag(P)^{-1}$. The entry-wise adaptive learning rates of AdamW is presented in $diag(P)$ and our method adds another level of control (by grouping the d entry-wise learning rates into K groups, each with a meta-learning rate that we design). In contrast, the traditional learning rate for Adam is one single meta-learning rate that lacks the degree of freedom.
>
> From the algorithmic viewpoint,  our meta-framework takes an optimizer as a black box, as presented in our algorithm 1. We only use some extra forward functions to decide the optimal learning rates given the parameter grouping.
>
> 3. Sorry we missed this. We kindly refer to Appendix A.6 for the experimental details of ViT classification. Please let us know if this is sufficient or if you have questions about the details of other experiments. We will release the code upon acceptance.
>
> 4. We did experiment on pretraining one regression and one classification tasks in section 4.3. However, we are limited in computational resource for further pretraining experiments, which we leave for future work.

---

> > ### Comment · Reviewer_dT3U · 2024-11-25
> >
> > I appreciate the authors' response to my questions. Some of my coments have been addressed, but I am still not convinced about the novelty and significance of this paper. Moreover, the algorithms that are compared with Hi-DLR, such as Prodigy and D-adaptation, are with theroetical guarantees (and if you check the papers that proposed Prodigy and D-adaptation, a significant proportion of the contents are to develop theoretical guarantees). Therefore, I feel that this paper still has significant disadvantages compared with existing works. Finally, I still feel that this paper needs a thorough revision before publication. I feel that the introduction section does not give a smooth logic, and descriptions/discussions about experiments are not very well written (for example, what do "Constant", "Linear decay", "Cosine decay" mean in Table 1, and what algortihm are these schedules referring to? I think this type of information has to be provided in the main paper).

---

> > > ### Author Response · Authors · 2024-11-25
> > >
> > > Thank you for your reply.
> > >
> > > *the algorithms that are compared with Hi-DLR, such as Prodigy and D-adaptation, are with theroetical guarantees*
> > >
> > > As we stated in the rebuttal and paper, methods like Prodigy and Dadaptation only work in ULR. Hence, they cannot solve the DLR problem as we proposed.
> > >
> > > *I feel that the introduction section does not give a smooth logic*
> > >
> > > Can you be more specific on where we can improve on this? Thanks.
> > >
> > > *descriptions/discussions about experiments are not very well written (for example, what do "Constant", "Linear decay", "Cosine decay" mean in Table 1, and what algortihm are these schedules referring to*
> > >
> > > These are heuristic learning rate schedulers that are widely used in DL. We added a sentence to introduce them in the main text. And we kindly provide the following links for your reference:
> > >
> > > "Constant": a constant learning rate.
> > >
> > > "Linear decay": https://pytorch.org/docs/stable/generated/torch.optim.lr_scheduler.LinearLR.html
> > >
> > > "Cosine decay": https://pytorch.org/docs/stable/generated/torch.optim.lr_scheduler.CosineAnnealingLR.html
> > >
> > > We'll put these links in the appendix if needed.
> > >
> > > We hope our answers can resolve some of your concerns.

---

> > > > ### Comment · Reviewer_dT3U · 2024-11-25
> > > >
> > > > Regarding the learning rate schedules, what I meant to ask was the base algorithm. Now I see that in the caption it was mentioned that the optimizer is AdamW so I have no more questions about it. However, I do feel that in general, additional paragraphs discussing and introducing some basic experiment setups may be very helpful.
> > > >
> > > > Regarding the introduction section, the current paper begins with several paragraphs that have bolded headings. This approach seems somewhat atypical to me. The opening sentences provide direct definitions of DLR, parameter group, and ULR, without offering any context or discussion. I believe it would be beneficial to improve this section by adding more context and a smoother introduction.
> > > >
> > > > Regarding your comment that "Prodigy and Dadaptation only work in ULR. Hence, they cannot solve the DLR problem as we proposed.", I find that it contradicts the comment at lines 77-78 "we can view the adaptive optimizers including Adam as SGD with coordinate-wise DLR". First of all, it is not very clear what do you mean by the "DLR problem". My understanding is that DLR is a concept highlighted in this paper that can potentially help improving efficiency, but I am not sure if it is appropriate to formulate any concrete "DLR problem". In addition, since Prodigy and Dadaptation have Adam versions, and Adam can be treated as coordinate-wise DLR, it does not seem straightforward that Prodigy and Dadaptation cannot solve the "DLR problem".

---

> > > > > ### Author Response · Authors · 2024-11-26
> > > > >
> > > > > We appreciate your feedback on the DLR narratives and have revised the presentation from the angle of "degree of freedom". Prodigy and Dadaptation on Adam are studying the DLR problem at the degree of freedom 1. However, our method can study the DLR problem at the degree of freedom that is larger than 1.
> > > > >
> > > > > To illustrate this briefly, we consider vanilla SGD and SignSGD (which is a special case of Adam) on 2 parameters.
> > > > >
> > > > > SGD(g): $w_t-w_{t+1}=\eta g=[\eta g_1,\eta g_2]$
> > > > >
> > > > > SignSGD(g): $w_t-w_{t+1}=\eta sign(g)=[\eta sign(g_1),\eta sign(g_2)]$
> > > > >
> > > > > In SignSGD, the learning rate is $\eta$, which applies to $sign(g)\in R^2$; equivalently, this SignSGD-ULR can be viewed as SGD-DLR, which applies $\eta_1:=\eta/|g_1|$ to $g_1$ and $\eta_2:=\eta/|g_2|$ to $g_2$. Hence SignSGD is a special case of SGD with coordinate-wise learning rates. However, the two learning rates $\eta_1,\eta_2$ are governed by one single hyperparameter $\eta$, meaning **the degree of freedom in hyperparameters is always 1** even in the DLR method. If we denote the degree of freedom in parenthesis, then Adam-ULR(1) is equivalent to SGD-DLR(1).
> > > > >
> > > > > In contrast, our formulation in Line 147, is studying SGD/Adam-DLR(K), with multiple degress of freedom in hyperparameters that Hi-DLR can give suggestion on (again, Prodigy/D-adaptation only gives suggestion on 1 $\eta$, hence degree of freedom is 1).
> > > > >
> > > > > We reformulate the definition of the "DLR problem" that we are actually solving in section 2.2. Please let us know if it is clear now.

---

### Official Review · Reviewer_DnhU · 2024-11-03

**Soundness:** 3
**Presentation:** 3
**Contribution:** 3
**Rating:** 5
**Confidence:** 3

**Summary:**

The paper introduces a novel method called Hessian-informed Differential Learning Rate (Hi-DLR) to optimize the training of neural networks by adapting learning rates based on the curvature of the loss function, which enhances convergence across various tasks. It highlights the limitations of existing Parameter-Efficient Tuning (PET) methods, demonstrating that no single PET method is universally effective, as performance varies significantly with different datasets and model architectures. The authors propose a flexible, model-agnostic meta-framework that adaptively selects the most effective PET methods and parameter groups based on their Per-Parameter Influence (PPI) during training.

**Strengths:**

The paper presents the Hessian-informed Differential Learning Rate (Hi-DLR) method, which enriches the approximation of Hessian information to leverage the varying loss curvature of different parameters through adaptive learning rates, enhancing the training efficiency of deep learning models .

The authors propose an efficient algorithm for computing Hi-DLR, incorporating a novel diagonalization technique that significantly reduces computational costs while effectively separating the contributions of different parameter groups, thus facilitating faster training without sacrificing performance .

The paper introduces a flexible, model-agnostic meta-framework for Parameter-Efficient Tuning (PET) that utilizes per-parameter influence to dynamically select trainable parameters. This adaptive approach allows for improved performance across various tasks and models, addressing the limitations of existing PET methods

**Weaknesses:**

1. This method includes other time-consuming steps and is incomplete without measurement. Under the same conditions, would a standard AdamW approach achieve a similar result?


2.The effectiveness of Hi-DLR depends on appropriate parameter grouping; suboptimal groupings can lead to less effective learning rate adjustments, potentially hindering performance. In the paper, parameter groups are adjusted manually, which requires explanation. Additionally, for large models, the hyperparameter K warrants further discussion.

3.When group number K is a large number, the update period will increase as its learning rate update every \phi iterations, which might influence the convergence.

**Questions:**

Could the authors provide the source code for reproduction?

---

> ### Author Response · Authors · 2024-11-22
>
> We thank the reviewer for the comments! We address them below and welcome more feedbacks. We would appreciate it if the reviewer could raise the score if satisfied.
>
> 1. *This method includes other time-consuming steps and is incomplete without measurement.*
>
> Thank you for mentioning this. We have added a detailed complexity analysis in Appendix B in our revised version.
>
> *Under the same conditions, would a standard AdamW approach achieve a similar result?*
>
> Empirically we have shown that a standard AdamW can achieve similar (but maybe worse) results in many tasks (e.g, table 1 and 2). However, we note that this only holds when AdamW is using a proper learning rate, which takes time to search or tune. Under the same condition (i.e. using K-group DLR), if we use grid search, the learning rate searching time for a standard AdamW would be $O(K^2)$. Our algorithm only takes $O(K/ \Phi)$ and can be further compressed to $O(1)$ if we choose $\Phi=O(k)$. Besides, Hi-DLR can analyze the parameter importance by PPI while using standard AdamW cannot achieve this.
>
> 2. *In the paper, parameter groups are adjusted manually, which requires explanation. Additionally, for large models, the hyperparameter K warrants further discussion.*
>
> Our main focus is given a grouping, how to design proper DLR for it, instead of how to design the grouping. In this work, we also explore the second question in two ways: (1) We chose application areas that the grouping strategies come naturally, such as multi-task learning, NAM and PET, e.g. PET naturally has two groups, the frozen one and the trainable one. (2) We construct a large enough set in Section 5 (say K=6, which gives $2^6$ sub-groupings) and use PPI to select the optimal sub-grouping. These two ways can extend to larger models without reconsidering K. For instance, in Table 7, we select the grouping on GPT-small and directly apply to GPT-large.
>
> We have added a new section to discuss our limitations and future directions in Appendix C.
>
>
> 3. *When group number K is a large number, the update period will increase as its learning rate update every \phi iterations, which might influence the convergence.*
>
> We agree the training time will increase as K becomes large, if we don't use other tricks. In our paper, we have experimented with K up to 40 in CelebA. For even larger K, one remediation is to use linearly larger $\Phi$ as we explained in Section 3 "When to derive".
>
> *Could the authors provide the source code for reproduction?*
> We will release the code upon acceptance.

---

> > ### Author Response · Authors · 2024-11-26
> >
> > Dear reviewer,
> >
> > We hope you are satisfied with our point-to-point response. Please kindly let us know whether we can improve in the last day of rebuttal. It would be greatly appreciated if you could consider raising the score.

---

### Official Review · Reviewer_sBXW · 2024-11-05

**Soundness:** 3
**Presentation:** 4
**Contribution:** 3
**Rating:** 6
**Confidence:** 3

**Summary:**

The main idea of this paper is to use different learning rates for different parameter groups. This is a common knowledge among optimizer designs that for different part of the model, e.g., the weight, bias, head, norm, etc., need different learning rates, however, how to search for the best learning rates remains open. This paper proposes to use the second-order approximate of the loss function to solve for the adaptive learning rates. In addition, the second-order approximation are done by regression on random sampled directions. The proposed algorithm significantly enhances training efficiency. Moreover, the paper introduces a per-parameter influence index, which identifies the most influential parameters, facilitating more efficient parameter-specific training.

**Strengths:**

Well-written, the intuition of this paper is clear, if the shape of a convex optimization surface is given, we can obtain the optimal learning rates for each directions directly. This can be done on top of any gradient momentum, and preconditioning regularizations.

Although the dimension of the parameters are large, the authors pick a few directions in the space (K directions) to get a low dimensional second-order approximation for learning rate searches.

The idea is simple but effective.

The authors also designed a per-parameter influence, that can differentiate the most influencial parameters for training. By freezing all other parameters under a PPI threshold, the authors can largely reduce the training cost of models, while preserving most of the testing accuracies.

**Weaknesses:**

Although everything diagonalized, the computational cost of the search is relatively high when fitting the second-order model, especially when $K$ is large.

It seems to me that the algorithm in principle is partly based on search, that you first move towards K directions defined by the split of K parameter groups, and than decides the curvature of each direction (fit a second order function) to get a learning rate on the parameter groups. I wonder if the algorithm can actually be designed more directly, for example instead of randomly sampled $\eta$s, fix several points on K lines, and than estimate curvature, make the learning rate on a line the negative curvature, etc.

The authors discussed the empirical comparison with other adaptive learning rate algorithms like prodigy and adaptation, etc. however, they did not discuss the relationship of their adaptation with previous adaptive algorithms in principle.

**Questions:**

$d_k$ is used both as an update direction, and as a notation of dimensionality in your paper.

In Equation (3.1), and when fitting the quadratic function, are you using the gradient calculations, or the corrected gradients like the momentum gradients in Adam? Does that mean second order Taylor expansion on any direction? It would be better if the authors could discuss how does their adaptive gradient interact with Adam's momentum, and adaptive schedule. Are they completely orthogonal, or how they influence each other?

For LoRA, the optimal learning rates ratios for matrix A and B can be fixed (Hayou et al., 2024) . How does your schedule interact with this ratio? If $\lambda$ is fixed, then your algorithm cannot work further on Lora+?

---

> ### Author Response · Authors · 2024-11-23
>
> We thank the reviewer for the comments! We address them below and welcome more feedbacks. We would appreciate it if the reviewer could raise the score if satisfied.
>
> *Although everything diagonalized, the computational cost of the search is relatively high when fitting the second-order model, especially when $K$ is large.*
>
> We have added a detailed complexity analysis in Appendix B. In a full-parameter training on a single GPU, the relative training time of Hi-DLR is $\frac{1}{1+\frac{4K}{3\Phi}}$. We agree the training time will increase as K becomes large, if we don't use other tricks. In our paper, we have experimented with K up to 40 in CelebA. For even larger K, one remediation is to use linearly larger  as we explained in Section 3 "When to derive".
>
> *I wonder if the algorithm can actually be designed more directly, for example instead of randomly sampled $\eta$s, fix several points on K lines, and than estimate curvature, make the learning rate on a line the negative curvature, etc.*
>
> Yes, there are some flexibility of designing the algorithm. We have tested fitting points $[-2,-1,1,2]*\eta_k$ for $k=1,...,K$ and the results are indistinguishable. We'll add this to algorithm 1.
>
>
> *The authors discussed the empirical comparison with other adaptive learning rate algorithms like prodigy and adaptation, etc. however, they did not discuss the relationship of their adaptation with previous adaptive algorithms in principle.*
>
> We have mentioned Prodigy and Dadaptation in paragraph "Automatic ULR" in line 105. These methods in their current form only work in ULR. Hence, they cannot solve the DLR problem as we proposed.
>
>
> *$d_k $ is used both as an update direction, and as a notation of dimensionality in your paper.*
>
> Sorry for the confusion. $d_k$ represents the number of parameters in group k but $\mathbf{d_k}$ (in bold) is an update direction. We'll change the notation in camera ready.
>
> *In Equation (3.1), and when fitting the quadratic function...*
>
> We use the corrected/pre-conditioned gradient when using Adam. Yes, the second order Taylor expansion is on any direction. We haven't studied the relationship but we know our adaptive gradient is positively correlated to the corrected gradient, regardless of Adam's momentum, and adaptive schedule. To see this, assume $[g_1, g_2]$ is the gradient of Adam grouped into two groups. Then Hi-DLR leads to $[\eta_1 g_1,\eta_2 g_2]$. It is obvious that the inner product $\eta_1 ||g_1||^2+\eta_2 ||g_2||^2>0$ which always holds because Hi-DLR gives positive learning rates.
>
>
> *For LoRA, the optimal learning rates ratios for matrix A and B can be fixed (Hayou et al., 2024) . How does your schedule interact with this ratio? If $\lambda$ is fixed, then your algorithm cannot work further on Lora+?*
>
> Our algorithm is compatible with Lora+. If the $\lambda$ is fixed, the optimal $\eta$ can be found by Hi-ULR because the task changes from finding optimal $[\eta_1,\eta_2]$ to $[1, \lambda]*\eta$.

---

### Note · Authors · 2025-01-20

I have read and agree with the venue's withdrawal policy on behalf of myself and my co-authors.